# Measuring and Mitigating Constraint Violations of In-Context Learning for Utterance-to-API Semantic Parsing

**Shufan Wang**[1][*], **Sébastien Jean**[2][†], **Sailik Sengupta**[2], **James Gung**[2],
**Nikolaos Pappas**[2], **Yi Zhang**[2]
[1]University of Massachusetts, Amherst    [2]AWS AI Labs
shufanwang@umass.edu, {sebjean, sailiks, gungj, nppappa, yizhngn}@amazon.com

## Abstract

In executable task-oriented semantic parsing, the system aims to translate users' utterances in natural language to machine-interpretable programs (API calls) that can be executed according to pre-defined API specifications. With the popularity of Large Language Models (LLMs), in-context learning offers a strong baseline for such scenarios, especially in data-limited regimes (Hu et al., 2022; Shin et al., 2021). However, LLMs are known to hallucinate and therefore pose a formidable challenge in constraining generated content (Parikh et al., 2020). Thus, it remains uncertain if LLMs can effectively perform task-oriented utterance-to-API generation where respecting API's structural and task-specific constraints is crucial. In this work, we seek to measure, analyze and mitigate such constraints violations. First, we identify the categories of various constraints in obtaining API-semantics from task-oriented utterances, and define fine-grained metrics that complement traditional ones. Second, we leverage these metrics to conduct a detailed error analysis of constraints violations seen in state-of-the-art LLMs, which motivates us to investigate two popular mitigation strategies– Semantic-Retrieval of Demonstrations (SRD) and API-aware Constrained Decoding (API-CD). Our experiments show that these strategies are effective at reducing constraints violations and improving the quality of the generated API calls, but require careful consideration given their implementation complexity and latency.

## 1 Introduction

In task-oriented dialog (TOD) settings, executable semantic parsers maps natural language utterances from human users to machine-interpretable meaning representations that help achieve users' goals. In order to effectively carry out the user's command, such task-oriented semantic parsers need

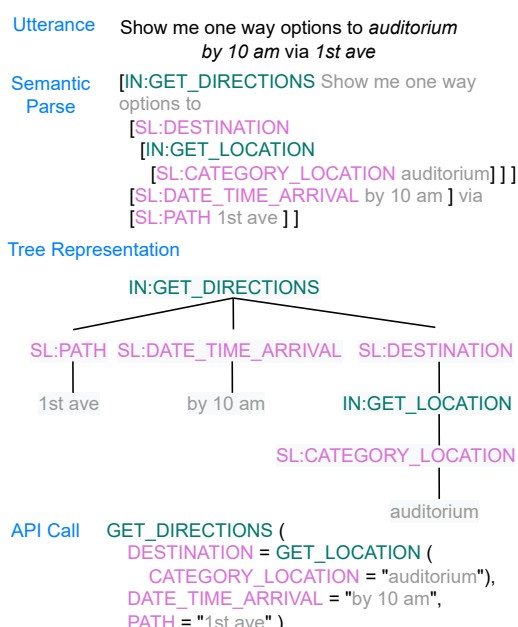

Figure 1: A nested example from the TOPv2 dataset consisting of an utterance, a corresponding semantic parse, the extracted tree representation and the corresponding API call. We measure & improve the ability of a Large Language Model (LLM) to generate the API call given an utterance with In-Context Learning (ICL).

to be able to interact with the real-world execution environment by translating users' utterances to API calls, that are executable according to pre-defined API documentation. For example, in Figure 1, given the user's utterance *Show me one way options to auditorium by 10am via 1st ave*, the executable task-oriented semantic parser should not only understand the semantic parse structures from the user's input, but more importantly, be able to generate the API call:

```
GET_DIRECTIONS   (   DESTINATION   =
GET_LOCATION ( CATEGORY_LOCATION =
"auditorium" ), . . ., PATH="1st ave") )
```

where the parsed intents and slots represent functions and arguments in the API calls.

Existing work has tackled task-oriented seman-

---

[*]Work performed during an internship at AWS AI Labs.
[†]Corresponding author

tic parsing by using auto-regressive language models (Rongali et al., 2020; Mansimov and Zhang, 2022; Hu et al., 2022; Shin and Van Durme, 2022). Recent advances of pre-trained Large Language Models (LLMs), especially those trained with code, present an exciting opportunity to frame executable semantic parsing tasks as an API call generation problem. In Hu et al. (2022), Shin et al. (2021) and Shin and Van Durme (2022), LLMs prompted with examples are able to generate code-like representations based on user queries. In-context Learning (ICL) (Brown et al., 2020) is a particularly appealing approach for semantic parsing tasks as it allows developers to (1) make use of LLM's transfer learning abilities in low-resource settings with only a small collection of labelled training data, and (2) easily prototype LLM-based systems without needing re-training or fine-tuning.

However, it has been shown that outputs from LLMs are prone to hallucination, lack controllability and have high sensitivity to prompt design (Parikh et al., 2020; Raunak et al., 2021; Rashkin et al., 2021), even when these models are prompted with human-written instructions and selected exemplars (Liu et al., 2022). These weaknesses are concerning for executable semantic parsing, as generated output needs to respect the task constraints specified by the API documents.

In this work, we investigate the question, *can LLMs with in-context learning respect the task-specific constraints required for task-oriented semantic parsing?* To the best of our knowledge, there is no existing work that categorizes and defines fine-grained task-specific constraints for task-oriented semantic parsing or systematically analyzes constraints violations in LLM outputs in the ICL scenario. We begin by identifying and categorizing the task constraints in Section 2, focusing on the TOP-V2 dataset (Chen et al., 2020), an existing task-oriented semantic parsing dataset, and detecting constraints violations in the output of GPT-NeoX (Black et al., 2022), a state-of-the-art LLM trained on code.

We observe that task-specific constraint violation are ubiquitous when using ICL in practice (ranging from $3 - 14.8\%$), while syntactic constraints are lower ($\approx 0.6\%$). In Section 3, we show that the proposed constraint violation metrics serve as a useful diagnostic tool, providing a fine-grained view and complementing existing evaluation measures in (Mansimov and Zhang, 2022). Our analysis of errors based on these constraint violation metrics reveal important failure modes, such as the lack of examples pertaining to a particular intent in the ICL input prompt, dependence on particular task-specific keywords, scenarios when hallucination is likely, etc. and inspires us to develop mitigation strategies that reduces constraint violations.

We investigate two popular mitigation strategies– retrieval-based augmentation (Pasupat et al., 2021; Gupta et al., 2022) and constrained decoding (Shin and Van Durme, 2022). In section 5, we show that our Semantic-Retrieval of Demonstrations (SRD), despite its implementation simplicity, improves the semantic parsing quality in the ICL setting and can also reduce constraint violation to an extent. On the other hand, when task-specific APIs are available, our API-aware Constrained Decoding (API-CD) strategy can completely eliminate constraint violations. An amalgamation of these two complementary approaches provides the best results, but incurs some tradeoffs in efficiency. We analyze this trade-off, offering advice for practitioners. Finally, we plan to release the modified TOPv2 dataset to encourage further research in this direction. In summary, our key contributions are:

(1) Defining and introducing constraint violation metrics for generated API calls (§2).
(2) Analysis of ICL failure modes using proposed metrics and proposal of mitigation strategies (§3§4).
(3) Showcasing efficacy of solutions to completely eliminate constraint violations (§5).

## 2 Defining Constraints Violations

Recent TOD systems, such as Shu et al. (2022) and Andreas et al. (2020), take (single/multi-turn) utterances of users as input and generate executable programs (*API calls*) to achieve user goals. Such systems are expected to not only generate effective programming codes but also respect the API constraints, relevant to the application domains, specified by developers. To study the task of generating valid API calls, we convert the popular TOD benchmark dataset TOPv2 (Chen et al., 2020) to an utterance-to-API format.[1] Specifically, TOPv2 consists of pairs of *user-utterance* and *semantic parse* representing the users' goals; we process the dataset by converting the semantic parses to API calls (see Figure 1).

---

[1]While the data in (Shu et al., 2022) would have been appropriate for our study, it is small in size and unreleased.

Note that the TOPv2 dataset contains many compositional/nested queries, where a particular slot value is the result of another intent. This is reflected under the dialog-to-API framework as nested calls, where the argument value to some function can be the return value from another function. Hence, an API call can be represented as a flattened list of functional calls, which we now define as a flattened list representation (examples for flattened list representations are in appendix A.1).

**Definition 2.1.** *A **flattened list representation** of an API call $y$ is a list of tuples $\{(f_i, \mathcal{A}_i)\}_{i=1,...n}$, where each $f_i$ is a* function *(corresponding to the* intent*) present in the API call and $\mathcal{A}_i$ is a set of $(a, v)$ pairs (corresponding to the* slots *and slot values). Each function $f_i$ is defined to be **associated** with all* arguments $a's$ *from $\mathcal{A}_i$.*

For compositional/nested API calls, the values $v$ can be other functions ($f$), while for non-compositional APIs they are resolvable/grounded values. It follows that in the latter case, the flattened list has only one function call.

## 2.1 Task-specific Constraints

We define four categories of constraints associated with the generation of API calls. The first constraint imposes a restriction on the overall output.

**Definition 2.2** (Structural Constraint $C_s$)**.** *An API call expression satisfies this constraint iff the expression can be parsed into a valid abstract syntax tree.*

The other constraints impose task-specific restrictions on the generated functions, the arguments, and the mutual interactions between them. Concretely, we first define a triple $(V_f, V_a, V_x)$, where (1) $V_f$ represents the set of valid functions ($f$); *(2) $V_a$ represents the set of valid arguments $a$, and* (3) $V_{fa}$ is the set of valid pairs of functions ($f$) and its associated arguments ($a$). We now define these task-specific constraints as follows.

**Definition 2.3** (Functional Constraint $C_f$)**.** *An API call $y$ violates $C_f$ if there is any function $f \in y$ and $f \notin V_f$.*

**Definition 2.4** (Argument Constraint $C_a$)**.** *An API call $y$ violates $C_a$ if there is any argument $a \in y$ and $a \notin V_a$.*

**Definition 2.5** (Function-argument Association Constraint $C_{fa}$)**.** *An API call $y$ violates $C_{fa}$ if there is any associated function-argument pair $(f, a) \in y$ and $(f, a) \notin V_{fa}$.*

To measure these constraints, we obtain the sets $V_f$, $V_a$, and $V_{fa}$ using the functions, arguments, and interactions in the entire training data of TOPv2. We note that one should use API-specific documentation, when available, to define these more efficiently. Further, we defined $V_f$ and $V_a$ as a set, and represented $V_{fa}$ as a dictionary.

In this work, we focus our attention to hard-constraint satisfaction, i.e. for every generated API call $y$, $C_*$ is either a $1$ or $0$. Thus, even if (say) $90\%$ of $y$ forms a valid abstract syntax tree, or $y$ has a function $f$ that is a synonym of function names $\in V_f$, it gets no brownie points. We feel this is reasonable given the high standards of API calls (e.g. even a single character typo in a function name results in compilation/run-time failures). Let us consider the metrics for the following example.

```
1  [User Utterance]
2  show my alarms for tomorrow
3
4  [Generated call]
5  SHOW_ALARMS ( DATE_TIME = "tomorrow")
```

$$
\begin{array}{rcl}
C_s & = & 1 \quad \text{✔ abstract syntax tree} \\
C_f & = & 0 \quad \text{✘ function name = GET\_ALARMS} \\
C_a & = & 1 \quad \text{✔ arg name (DATE\_TIME), ✔ value (tomorrow)} \\
C_{fa} & = & 0 \quad \text{✘ incorrect function name implies no} \\
& & \quad \text{valid function-argument pair} \in V_{fa}
\end{array}
$$

Note how each individual input-output provides a distinct constraint violation signature. To evaluate a system's success at constraint satisfaction, we compute a constraint violation rate, i.e. the percentage of constraint violations for each constraint violation category over all the test examples. While existing metrics like exact-match, F1, and accuracy give a top-level view of the model's capability for dialog-to-API scenarios, our metrics can identify more granular aspects; thereby giving a more complete picture. For example, a model scoring low on exact-match may still be great at correcting identifying functions and argument names from few-shot data in prompts and simply be bad at structural constraints. In addition, the fine-grained diagnosis can help us design better solutions.

## 3 Analyzing Constraint Violation

As we noted above, the use of exact match and F1-scores (Mansimov and Zhang, 2022) can only provide a bird's eye view of the model's failures. Unfortunately, recently employed metrics like BLEU and Execution Match Ratio (EMR) for such scenarios (Shu et al., 2022) are also incapable of providing a magnified view. Further, these metrics can

| Reasons for violation | Constraint Violation Examples | Constraint Violation Analysis |
|---|---|---|
| Misunderstanding task semantics

violates $C_f, C_a, C_{fa}$ | **[Utterance]:** show me wrecks to avoid on the interstate
**[Ground-truth]:**
**GET_INFO_TRAFFIC** ( OBSTRUCTION_AVOID = " wrecks " , LOCATION = GET_LOCATION ( CATEGORY_LOCATION = " the interstate " ) )
**[Prediction]:**
**GET_INFO_WRECKS** ( LOCATION = " the interstate" ) | The system simply uses the template GET_INFO_* that is ubiquitous in the training data (and therefore in the prompt). Although it has seen GET_INFO_TRAFFIC in the prompts, it fails to make the semantic implicature from *wrecks* to *traffic* during generation, thereby resulting in an undefined function call GET_INFO_WRECKS and violating all three task-specific constraints. |
| Misunderstanding slots/arguments associated with a task

violates $C_{fa}$ | **[Utterance]:** 10 Day forecast please.
**[Ground-truth]:**
GET_WEATHER ( **DATE_TIME** = " 10 Day " )
**[Prediction]:**
GET_WEATHER ( **DATE_TIME** = " today ", **FORECAST_DAYS** = " 10 " ) | The system fails to realize that DATE_TIME can support 10 Day as an argument value. Hence, it hallucinates a new argument FORECAST_DAYS to provide this information. Note that its argument definition for DATE_TIME is correct, and thus it only violates $C_{fa}$. |
| Lack of relevant examples in the prompt

violates $C_f, C_{fa}$ | **[Utterance]:** show my alarms for tomorrow
**[Ground-truth]:**
**GET_ALARM** ( ALARM_NAME = GET_TIME ( DATE_TIME = " for tomorrow " ) )
**[Prediction]:**
**SHOW_ALARMS** ( DATE_TIME = " tomorrow " ) | The GET_ALARM has a small support in the training data. Hence, sampling omits it in the example prompt. The model, oblivious to this function call, violates $C_f$ by generating an undefined function SHOW_ALARMS. |
| Inability to handle compositionality

violates $C_a, C_{fa}$ | **[Utterance]:** add reminder feed sparky 6pm daily
**[Ground-truth]:**
CREATE_REMINDER ( TODO = " feed sparky " , **RECURRING_DATE_TIME** = **GET_RECURRING_DATE_TIME** ( **DATE_TIME** = " 6 pm " , FREQUENCY = " daily " ) )
**[Prediction]:**
CREATE_REMINDER ( TODO = " feed sparky ", **PERSON_REMINDED** = " me ", FREQUENCY = " daily ", **AMOUNT** = " 6pm " ) | Correctly returning the value to RECURRING_DATE_TIME requires the execution of a nested function call GET_RECURRING_DATE_TIME with the provided time. The models fails to comprehend this nested call and flattens the information required by hallucinating arguments, such as AMOUNT and PERSON_REMINDED, associated with the primary function call, thereby violating $C_a$ and $C_{fa}$. |

Table 1: Error analysis of constraint violations made by the GPT-NeoX model on the TOPv2 dataset with ICL.

often be difficult to interpret in dialog-to-API scenarios. Hence, these metrics cannot help guide us to analyze particular test examples for insights in task-specific constraint violations, such as the lack of meaningful in-context examples, and the models' ability to comprehend general API definitions and process compositionality. In this section, we first perform the exercise of analysing constraint violations guided by our constraint violation metrics. Second, we leverage our analysis to hypothesize reasons for these violations.

In Table 1, we highlight a few example violations along with the unique signature generated by our metrics and analyze the common reasons for such constraint violations.

**Misunderstanding APIs and task semantics** In the first row, we see an example where the generated API call violates all of the task-specific constraints. When we see high counts of this for a particular domain (eg. traffic) in TOPv2, we can quickly infer that the system *fails to fully comprehend the task semantics* and is hallucinating function names, argument names, and making up associations between these hallucinated functions and arguments. In such scenarios, we often observe that the model is unable to make semantic jumps

from words in test utterances to existing function and arguments. While this could be due to lack of diverse vocabulary usage in the ICL prompts, enumerating all such semantic mappings in an example prompt may be unreasonable. In the second row, we observe that with the limited ICL examples, the model is unable to comprehend the domain of an argument/slot, but realizes the value '10 day' is important. Thus, it hallucinates a new argument and provides this value to it regardless of actual association constraints. This motivates us to consider constrained decoding strategies that restrict the generation space of function and argument names thereby limiting hallucinations.

**Lack of meaningful in-context examples** In the third row, we notice a scenario that has more to do with the ICL paradigm as opposed to shortcoming of the LLM itself. While the model can learn from in-context examples, lack of certain function (or argument) names in the prompts can derail the model to come up with incorrect (although semantically similar) function names. This motivates us to consider demonstration retrieval strategies.

**Inability to handle compositionality** In the final row, we highlight a failure mode that is ubiquitous in generated API calls across all domains

of TOPv2. Specifically, when dealing with compositional queries (nested functional calls), the generated API calls tend to flatten the call structure and bypass necessary nested API calls to correctly fulfil these function queries. In turn, these violate several task-specific API constraints. We further highlight these in our experiments (§5).

## 4 Mitigation Strategies

We study two popular mitigation strategies to reduce constraint violations– (1) a semantic retrieval approach for in-context example selection that improves the prompt design, and (2) an API-aware constrained decoding strategy that constrains token generation based on API information.

### 4.1 Semantic Retrieval of Demonstrations

Under the in-context learning approach, language models takes a list of input-output pairs in the prompt, referred to as *in-context examples* or *demonstrations*. In the context of our API generation task, such demonstrations consist of pairs of an utterance and corresponding ground-truth API calls $(u, a)$. In the examples shown in §3, these example demonstrations are randomly sampled from the same domain in the TOPv2 dataset and result in task-specific constraint violations. In this method, the key is to leverage the user's test utterance to strategically select the in-context demonstrations in the ICL prompt.

Consider user's utterance $u_t$ from the test dataset. We use a dense retriever model to obtain an embedding for the text $u_t$. We then retrieve top-k $(u_i, a_i)$ pairs from the training set based on the cosine similarity distance between $u_i$ and $u_t$ and arrange then in descending order to obtain the demonstrations for our prompt. For the dense retriever model, we leverage SBERT that has been trained on one billion sentence pairs using a contrastive learning objective (Reimers and Gurevych, 2019).

### 4.1.1 API-aware Constrained Decoding

In §2, we use the sets $V_f$, $V_a$, and $V_{fa}$ to define the set of possible values for function, arguments and associations for a particular domain. Looking at the hallucinations during our analysis in §3, adapting the model's generation capabilities to adhere to the API-specification is a proactive approach that can reduce constraint violations. We explore this direction in API-aware constrained decoding.

Previous works have proposed a constrained decoding algorithm that generates a "canonical form" of meaning representation (instead of natural language or programming code) (Shin et al., 2021; Shin and Van Durme, 2022). The canonical form is then mapped to semantic parse expressions associated with a pre-defined set of grammars written by humans. As we aim to generate API calls, instead of generating outputs that are then post-processed by human-written grammatical rules, we simply constrain the model to generate valid expressions according to the API constraints. For example, when the model generates the API call for a function $x$, we restrict the decoding algorithm to only generate arguments $a$ tokens that are the dictionary indexed with $x$, $V_{fa}[x]$. We can also ensure we generate well-formed API calls by enforcing syntactic constraints (eg. parenthesis closure) and other task-specific constraints (function and arguments names). A minor detail is that the function and arguments names are often tokenized (by the pre-trained model's tokenizer) to a sequence of sub-tokens; thus, we need to, define all the valid sequences of sub-tokens that form a valid function's (or argument's) name. For this purpose, we use a dynamic programming approach with backtracking to extract all such possible sequences and restrict the LLMs generation according to the extracted sub-token sequences.

## 5 Experiments

Using the constraints defined in the previous section, we now benchmark the performance of in-context learning with large language models on constraint satisfaction. In addition, we use the proposed mitigation strategies, namely retrieval-based and constrained-decoding-based, and show their effectiveness in reducing the constraint violation rates compared to the baselines. Finally, we discuss important practical considerations that accompany the proposed mitigation strategies.

**Large Language Model (LLM)**   To understand how well LLMs perform at generating API representation for utterances in the TOPv2 datasets, we use the GPT-NeoX model (Black et al., 2022), which is a 20B parameter model trained on the Pile data consisting of $\approx$ 825GB of raw text data (Gao et al., 2020). Given the training data contains textual sources (eg. books, subtitles, emails, Wikipedia, etc.), and API/code like data (eg. GitHub, stack exchange, hacker news), we find it befitting for our context. Other suitable LLM options for our setting were either under behind a

| Sampler | Query Types | Constraints Violations Metrics (%) ($\downarrow$) | | | | Semantic Parsing Metrics (%)($\uparrow$) | | |
|---------|-------------|------|------|------|----------|-------------|----------|--------|
| | | $C_s$ | $C_f$ | $C_a$ | $C_{fa}$ | Exact Match | Intent F1 | Slot F1 |
| Random | Flat | 0.63 | 7.6 | 2.77 | 9.6 | 34.7 | 75.4 | 45.3 |
| | Compositional | 0.67 | 12.8 | 3.36 | 15.4 | 7.0 | 61.8 | 34.8 |
| | All | 0.64 | 9.8 | 3.02 | 12.1 | 22.9 | 68.0 | 39.4 |
| SPIS-1 | All | 0.00 | 14.8 | 3.00 | 17.8 | 24.0 | 68.7 | 43.9 |

Table 2: Constraint violations metrics using ICL with GPT-NeoX on the TopV2 dataset. Using a SPIS-1 sampling strategy (that ensures coverage of all functions and arguments) in the ICL examples improves semantic parsing metrics, but degrades constraint violation metrics (rows 3, 4). Also, models commit more constraint violations when dealing with compositionality (rows 1, 2).

paywall (eg. OpenAI Codex) or smaller in scale (eg. GPT-Neo-2.7B, GPT-J-6B). [2]

**Dataset** To study the constraint violations from generating API calls in task-oriented settings, we use the TOD Semantic Parsing dataset TOPv2 (Chen et al., 2020). The TOPv2 dataset contains pairs of utterances and aligned semantic representations for a diverse set of source domains [3]. The semantic representation can either be flat or compositional in nature. For low-resource setting, the TOPv2 dataset restricts the training data using the *samples per intent and slot label (SPIS)* strategy (lower $\propto$ scarce) in two domains– Reminder and Weather. For completeness of evaluation, we use the SPIS sampling method and obtain low-resource training datasets for all domains in TOPv2 (and will release the data for future research). For the test datasets, we sample 200 examples from the test split for each of the eight domains in TOPv2 (including 100 flat queries and 100 compositional queries in each domain).

**Task Prompt Creation** The *in-context learning* approach allows language models to output predictions based on carefully constructed natural language text inputs (*prompts*). The prompt contains a task description, followed by input-output demonstrations, and finally the *test input*, for which an output prediction is desired. We then ask the language model to take the prompt as input and output API calls for the test query by greedy decoding. While our task descriptions and test inputs remain consistent across prompts, we consider different sampling strategies for composing in-context examples in the experiments below.

[2]We run GPT-NeoX on a single machine with 8 V-100 GPUs, each with 16GB memory, and take 18 hours to complete the in-context learning inference for our test examples.

[3]alarm, event, messaging, music, navigation, reminder, timer, and weather

## 5.1 Constraint Violation with LLMs

In these experiments, we use two sampling strategies– random and SPIS-1 sampling– to sample 10 in-context examples from a domain in the ICL prompt followed by a test utterance from the same domain. The SPIS-1 sampling ensures that every function and argument is represented at least once in the in-context examples, whereas the random sampler selects examples randomly from the training data. We construct the ICL prompt by concatenating the task descriptions, the in-context examples and the test query. For example, a prompt for the test query *Driving directions to the Eagles game* (from the navigation domain) looks as follows:

```
1  #[TASK DESCRIPTION]
2  Follow the examples below and generate API Calls
      from the users' utterances
3
4  #[IN-CONTEXT EXAMPLES]
5  Example 1:
6  User: What's traffic going to be like on Saturday
7  API Call: GET_INFO_TRAFFIC ( DATE_TIME = "on
      Saturday" )
8
9  Example 2:
10 User: How long is the flight from Fort Lauderdale
      to Jamaica
11 API Call: GET_ESTIMATED_DURATION ( METHOD_TRAVEL =
      "flight", SOURCE = "Fort Lauderdale",
      DESTINATION = "Jamaica" )
12
13 ... [10 examples] ...
14 #[TEST QUERY]
15 Example 11:
16 User: Driving directions to the Eagles game
17 API Call:
```

Note that for this baseline, we do not consider any mitigation strategies. In Table 2, we evaluate the corresponding output API call using the constraint violation metrics and traditional semantic parsing metrics.

In line with our observation in §3, we observe a higher rate of constraints violation for compositional queries in the first two rows of Table 2. In this case, the semantic parsing metrics offer a similar conclusion, showcasing the poor performance of the model on nested queries compared to flat

| Sampler | | Constraints Violations Metrics (%) ($\downarrow$) | | | | Semantic Parsing Metrics (%)($\uparrow$) | | |
|---|---|---|---|---|---|---|---|---|
| | | $C_s$ | $C_f$ | $C_a$ | $C_{fa}$ | Exact Match | Intent F1 | Slot F1 |
| Random | | 0.64 | 9.80 | 3.02 | 12.10 | 22.9 | 68.0 | 39.4 |
| SRD (ours) | +SPIS-5 | **0.00** | 3.49 | 1.00 | 6.20 | 37.7 | 81.4 | 52.2 |
| | +SPIS-10 | 0.20 | 2.60 | **0.70** | 4.84 | 40.6 | 82.4 | 55.7 |
| | +SPIS-25 | 0.10 | 1.84 | 1.10 | 4.34 | 43.2 | 86.0 | 58.9 |
| | +SPIS-50 | 0.10 | **1.10** | 1.00 | **2.88** | 44.4 | 86.7 | 60.4 |
| | +SPIS-100 | 0.40 | 1.20 | 0.89 | 2.96 | **49.6** | **88.2** | **63.5** |

Table 3: Applying a query-based Semantic Retrieval for Demonstrations (SRD) reduces both the constraint violation rates and improves the semantic parsing metrics. Increasing the training data subset from which SRD samples in-context examples improves results further.

queries.

In contrast, observing the last two rows of the table (where we consider *All* query types); it shows the impact of different samplers, and therefore different set of in-context examples, on the in-context learning performance. Given SPIS-1's more comprehensive coverage of functions and arguments compared to the random sampler, we observe higher semantic parsing metrics (EMs, intent and slot F1s). However, this higher coverage does not necessarily equate to a lower constraint violations rate– we notice a higher violation rate for $C_f$ and $C_{fa}$. Not only does this highlight the added diagnostic value our metrics offer complementary to semantic parsing metrics, but it also showcases that SPIS-1 sampling may not be sufficient to eliminate function name hallucination errors (see example in the third row of Table 1).

### 5.2 Efficacy of Mitigation Strategies

**Semantic Retrieval of Demonstrations (SRD)** Table 3 shows that by simply applying semantic retrievals to obtain top-10 demonstrations examples, we can reduce the constraint violations significantly. We cannot expect the availability of the full training data in a few-shot setting, as that is often the motivation for using ICL. Thus, we sample the top-10 demonstrations for semantic retrieval from a pool of SPIS sampled training set. While larger and more comprehensive training sets (higher SPIS) increase the effectiveness of our retrieval strategy in reducing constraint violations, we see gains even in the most extreme few-shot setting (SPIS-5). In addition to reducing constraint violations, the semantic retrieval is a simple strategy to implement that only augments the input prompt. Further, it also leads to better performance according to the traditional semantic parsing metrics. However, we also observe that the reductions in constraint vio-

lations with retrieval strategies start to reduce in magnitude after SPIS-50 although the increase in exact match continues at a steady pace up to SPIS-100. This indicates that constraints violations are more difficult to eliminate and necessitates the need for other mitigation strategies.

**API-aware Constrained Decoding (API-CD)** As shown in Table 4, our algorithm is able to eliminate all categories of constraint violations and improve on semantic parsing metrics (although the magnitude of improvement is disproportionately less for the latter metrics). One may ask how can constraint violation metrics be zero while semantic accuracy metrics are only a little better. For this, consider an example where the generated function name belongs to $V_f$ but is not the correct function name as per the input text. In addition, we find that the constrained decoding algorithm is 20% slower than original decoding as it needs to extract at run-time the appropriate sub-sequences that form the well-formed function and/or argument names. Improving its efficiency can be a promising future direction.

**Combined Approaches (SRD + API-CD)** Note that we can combine our proposed methods seamlessly as they are designed for different phases of ICL – (1) SRD for prompt generation and, (2) API-CD for decoding. In the last row of Table 4, we observe that this approach yields the best results, where SRD primarily improves the Semantic Parsing Metrics (SPM) and API-CD improves the Constraint Violation Metrics. Interestingly, the cases where API-CD improves the SPM are a subset of the cases where SRD improves SPM. Thus, we see the same SPM numbers as the SPIS-5 row of Table 3.

For practitioners, we note that implementing SRD is both simple and incurs little latency com-

| Sampling | Decoding | Constraints Violations Metrics (%) (↓) | | | | Semantic Parsing Metrics (%)(↑) | | |
|---|---|---|---|---|---|---|---|---|
| | | $C_s$ | $C_f$ | $C_a$ | $C_{fa}$ | Exact Match | Intent F1 | Slot F1 |
| Random | Original | 0.64 | 9.80 | 3.02 | 12.10 | 22.9 | 68.0 | 39.4 |
| | API-CD (ours) | **0.00** | **0.00** | **0.00** | **0.00** | **24.2** | **71.7** | **39.5** |
| SRD + SPIS-5 (ours) | API-CD (ours) | **0.00** | **0.00** | **0.00** | **0.00** | **37.7** | **81.4** | **52.2** |

Table 4: Applying API-aware Constrained Decoding (API-CD) eliminates all four categories of constraint violations (constraint violation rate = 0%) and improves the semantic parsing metrics. Combining both the semantic retrieval and constrained decoding reaps complementary benefits.

pared to random sampling. On the other hand, API-CD requires the API documentation to be fully available and can increase the latency costs by $\approx 20\%$ compared to the original decoding strategy. In the future, one can consider a tighter integration, where the knowledge obtained during the SRD phrase reduces the decoding output space during API-CD, thereby offering a speed-up. Unfortunately, the development complexity may outweigh the latency cost benefits (and some semantic parsing metric improvement) in this case.

# 6 Related Work

Generating semantic representation from natural language is a beast with many heads (Zettlemoyer and Collins, 2012; Berant and Liang, 2014; Dong and Lapata, 2018; Wang et al., 2019). In our work, we look at a subclass of this problem where the input is a task-oriented utterance and the output takes the form of API call representations. Our input format aligns us closely to work in task-oriented semantic parsing (Gupta et al., 2018), task-oriented dialog (Mansimov and Zhang, 2022; Xuan, 2020), and intent classification and slot filling (Aghajanyan et al., 2020; Weld et al., 2022), while our output representation invites challenges similar to ones seen in generating executable semantic parses (Liang, 2016; Zhong et al., 2020), database queries (Berant and Liang, 2014; Li and Jagadish, 2014; Zhang et al., 2019; Yu et al., 2020; Choi et al., 2021), and AWS API calls (Shu et al., 2022). Our problem borrows the API-styled output, similar to Shu et al. (2022), but does not rely on an entire conversation or dialog as the input medium. To this extent, we enhance the existing TOPv2 dataset (Chen et al., 2020), leveraged in task-oriented settings, with API-styled output representations.

With the recent popularity of Large Language Models, adapting them for various tasks with the abilities to use different tools is becoming ubiquitous. Qin et al. (2023) fine-tunes language models

to use tools with external APIs to execute complex instructions. Schick et al. (2023) demonstrates that language models can learn to use external tools via simple APIs in a self-supervised way by in-context learning. Similar to our setup, the in-context learning approach (Brown et al., 2020) has been considered for intent and slot filing (Yu et al., 2021), semantic parsing (Shin et al., 2021), by predicting English-like canonical representations first (Shin et al., 2021), meaning representations directly (Shin and Van Durme, 2022), and dialog-to-API output (Shu et al., 2022). None of these works consider, like us, fine-grained constraint violation metrics that can help to decipher the failure modes of ICL in their settings. Hence, benefits of in-context example retrieval do not show the shortcomings of these approaches in mitigating constraint violations (Shin et al., 2021; Liu et al., 2022; Rubin et al., 2022). While lexically constrained decoding has been beneficial in a wide variety of setting such as image captioning, machine translation, semantic parsing, and paraphrase generation (Anderson et al., 2017; Post and Vilar, 2018; Hu et al., 2019; Li et al., 2020; Lertvittayakumjorn et al., 2021a; Shin and Van Durme, 2022), dynamic adaptation of the generation lexicon by leveraging API definitions has not been investigated before.

# 7 Conclusion

In this paper, we consider the problem of generating an API-call-styled representation given an input task-oriented utterance using in-context learning (ICL) approaches with Large Language Models. We propose a set of fine-grained constraint violation metrics that align with the API specifications and show that it lets us diagnose fine-grained failure modes. Our analysis leads us to investigate two simple mitigation strategies than can be used alongside the ICL approach. A query-based Semantic Retrieval of Demonstration in ICL prompts helps obtain gains on traditional metrics while API-aware

constraint decoding helps eliminate the proposed constraint violations. Coupling the two approaches with ICL yields the best of both worlds. Future works may focus on speeding up the mitigation approaches, and finding solutions that couple the approaches in a tighter fashion.

## Limitations

**Focus on the in-context learning approach**  We explored generating API representations with the in-context learning approach due to its wide appeals to practitioners in the low-resource settings. The in-context prompts allow the model to make use of examples in the low-resource settings, without the need to update model parameters. However, our investigation in mitigating constraint violations is limited because when the practitioners are able to pre-train large language models, there may be other approaches, such as incorporating the constraints in the language model pre-training stage, that may help reduce constraint violations.

**Focus on "hard" constraints**  Additionally, in this work, we focus on categorical constraints (e.g. structural constraints, function constraints, argument constraints ...), which can be automatically detected without any ambiguity by checking the generated API calls against pre-defined dictionaries that stipulate the constraints according to API documents. Other forms of constraints, such as knowledge-driven constraints (Lertvittayakumjorn et al., 2021b), which requires common-sense reasoning, are not included in our study. Future works may investigate detecting and mitigation of constraints that require external knowledge.

## Ethical Considerations

Our study makes use of the GPT-NeoX model, a large pre-trained language model trained on collections of internet text. Large language models like GPT-NeoX are known to generate text that may reflect and spread biases from their training data. Hence, we advise post-processing on the generated outputs to remove generated content that is potentially offensive. Additionally, our constrained decoding strategy limits the output space and may therefore reduce the risk of using these language models.

## Acknowledgements

We thank the AWS AI Lab team members for the feedback and discussion on the paper, and the anonymous reviewers for their helpful comments.

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

# A Appendix

## A.1 Flattened API Calls

Given an API call, we flatten it to a list of function calls, where each function is represented by a function name $f$ and a set of argument-value pairs $\mathcal{A}$ (Table 5).

| Users' Utterances Ground-Truth API Calls | Flattened List Representation of the Ground-Truth API calls |
|---|---|
| **[Utterance]:** 
 add reminder feed sparky 6pm daily 

 **[Ground-truth]:** 
 CREATE_REMINDER ( TODO = "feed sparky", RECURRING_DATE_TIME = GET_RECURRING_DATE_TIME ( DATE_TIME = "6 pm" , FREQUENCY = "daily" ) ) | [ 
 $(f_1$=CREATE_REMINDER, 
 $\mathcal{A}_1$={ TODO: "feed sparky", RECURRING_DATE_TIME:GET_RECURRING_DATE_TIME }), 
 $(f_2$=GET_RECURRING_DATE_TIME, 
 $\mathcal{A}_2$={ DATE_TIME: "6 pm", FREQUENCY: "daily" }), 
 ] |
| **[Utterance]:** 
 What's traffic going to be like on Saturday 

 **[Ground-truth]:** 
 GET_INFO_TRAFFIC ( DATE_TIME = "on Saturday") | [ 
 $(f_1$=GET_INFO_TRAFFIC, 
 $\mathcal{A}_1$={ DATE_TIME: "on Saturday" }) 
 ] |

Table 5: Examples of API calls and their flattened list representations