# OpenReview forum: "Measuring and Mitigating Constraint Violations of In-Context Learning for Utterance-to-API Semantic Parsing"
_EMNLP/2023/Conference — EMNLP 2023 Findings_

### Official Review · Reviewer_dJAL · 2023-08-04

**Typos Grammar Style And Presentation Improvements:** Paper reads well.
**Soundness:** 4

**Excitement:**

3: Ambivalent: It has merits (e.g., it reports state-of-the-art results, the idea is nice), but there are key weaknesses (e.g., it describes incremental work), and it can significantly benefit from another round of revision. However, I won't object to accepting it if my co-reviewers champion it.

**Missing References:**

The related work looks solid.

**Paper Topic And Main Contributions:**

The authors present a set of improved metrics for measuring constraint violations in the domain of natural language to API semantic parsing. The research explores the use of these metrics within the context of in-context learning with large Language Models (LLMs). The main objective is to generate APIs that effectively solve tasks based on a given test natural language utterance and a limited set of training examples with ground-truth data.

The paper's contribution lies in its novel approach to evaluating constraint violations, which is crucial for refining the performance of natural language to API semantic parsing systems. By employing these metrics, the authors aim to enhance the accuracy and reliability of the API generation process.

The study introduces two potential solutions for mitigating constraint violations. Firstly, the authors suggest selecting in-context examples that closely align with the test utterance. This approach helps ensure that the generated APIs are relevant to the specific context and leads to more accurate results. Secondly, they propose the use of constrained decoding to guide the LLM generation towards producing valid API names and arguments. By imposing constraints during the decoding process, the authors anticipate an improvement in the adherence of generated APIs to the desired constraints.

To validate their proposed metrics and mitigation strategies, the authors conduct experiments on the TOPv2 benchmarks, a dataset introduced by Chen et al. in 2020. The results showcase the practical utility of the proposed metrics and the effectiveness of the mitigation techniques in real-world scenarios.

**Questions For The Authors:**

1. Why is the task of creating API calls different than the semantic parse task proposed in Chen et al. 2020. The can be deterministic transfer between the API calls and the semantic parse?

2. Line 130, release the modified TOPv2 dataset - What are the modifications?

3. Are the API definitions same for the entire trainings/test set ? If that is the case, why can't the constrained decoding techniques for semantic parsing not applicable here?

**Reasons To Accept:**

1. The paper introduces a set of metrics to assess constraint violations in the context of API semantic parsing, which includes API names, API arguments, and valid arguments within an API. These metrics prove to be highly valuable in understanding the performance of API semantic parsing systems. By quantifying constraint violations, the proposed metrics offer crucial insights into the effectiveness of the parsing process.

2.  The paper presents two mitigation techniques to address these constraints. Firstly, the authors suggest fetching similar queries from the training set, which helps in reducing constraint violations. This approach leverages contextually relevant examples to improve API generation accuracy. Secondly, the paper proposes constrained decoding as a means to further mitigate constraint violations. This technique guides the generation process, leading to APIs that conform more closely to desired constraints. The effectiveness of these mitigation strategies validates the utility of the proposed metrics.

3. The experimental results presented in the paper support the proposed metrics and mitigation techniques.

**Reasons To Reject:**

1. The lack of new ideas for the mitigation strategies to remove the constraint violations. -The role of selecting relevant information for in-context learning is popular in the retrieval augmented generation paradigm. Constrained decoding is also a popular technique to reduce parsing errors in semantic parsing literature.


**Reproducibility:**

5: Could easily reproduce the results.

**Reviewer Confidence:**

4: Quite sure. I tried to check the important points carefully. It's unlikely, though conceivable, that I missed something that should affect my ratings.

---

> ### Author Rebuttal · Authors · 2023-08-29
>
> We thank reviewer dJAL for their insightful reviews and for recognizing the value of our defined metrics on measuring constraints violations.
>
> > Weakness 1: The lack of new ideas for the mitigation strategies
>
> We agree with reviewer dJAL that the investigated strategies do not introduce novelty. However, we clarify that we do not claim novelty on the mitigation strategies. Our work focuses on the following aspects.
>  1. We formally define task constraints and constraint violations in the utterance-to-API generation task;
>  2. We evaluate existing LLMs’ ability to satisfy task constraints and conduct a human analysis on the causes for constraint violations;
>  3. We investigate mitigation strategies chosen based on our error analysis.
>
> Even though the investigated strategies are not novel, we hope that future research can draw insights from Section 5, where we compare and offer advice on using these strategies in practice.
>
> > Question 1: Why is the task of creating API calls different than the semantic parse task proposed in Chen et al. 2020. The can be deterministic transfer between the API calls and the semantic parse?
>
> Reviewer dJAL is correct that generating API calls is indeed equivalent to the executable semantic parsing task. Nonetheless we adopt the API call generation format for the following reasons:
> 1. Given the surge in interest in “executable” semantic parsing (where pre-existing APIs are used to ground the generated output), the utterance-to-API format can make our definition on constraints and evaluation more widely applicable to approaches that operate on pre-existing APIs (e.g. tool-augmented LLMs)
> 2. The utterance-to-API format (generating executable code based on natural language input) aligns more closely with the pre-training of state-of-the-art LLMs used, which are trained with code.
>
>
> > Question 2: Line 130, release the modified TOPv2 dataset - What are the modifications?
>
> We modify the datasets by mapping intents and slots in the semantic parse representation to the corresponding functions and arguments in the API call representation. An example is provided in Fig 1, where a “semantic parse” (second row) can be modified to an “API call” (last row).
>
>
> > Question 3: Are the API definitions same for the entire trainings/test set ? If that is the case, why can't the constrained decoding techniques for semantic parsing not applicable here?
>
> Yes, given a domain, the API definitions used remain consistent for the training / test set. We acknowledge that our constrained decoding strategy is indeed modified from constrained decoding techniques for semantic parsing, to satisfy pre-defined function and argument API constraints.

---

### Official Review · Reviewer_13ew · 2023-08-04

**Typos Grammar Style And Presentation Improvements:** NA
**Soundness:** 3

**Excitement:**

3: Ambivalent: It has merits (e.g., it reports state-of-the-art results, the idea is nice), but there are key weaknesses (e.g., it describes incremental work), and it can significantly benefit from another round of revision. However, I won't object to accepting it if my co-reviewers champion it.

**Missing References:**

NA

**Paper Topic And Main Contributions:**

Executable task oriented semantic parsing is used to convert user's natural language utterances to executable API calls. LLMs can provide a strong baseline for this task but they have a tendency to hallucinate, posing a significant challenge to generating constrained content. This work is related to measuring, analyzing and mitigating the constraint violations while using ICL with LLMs. Authors propose to identify the categories of various constraints in obtaining API semantics from task oriented utterances along with defining fine-grained metrics. These metrics are used for error analysis of constraint violations by LLM. Finally, two popular mitigation strategies are investigated which are SRD and API-CD. This work experimentally shows that these strategies help reduce the constraint violations and improve the quality of generated API calls.

**Questions For The Authors:**

See weakness section

**Reasons To Accept:**

1. This work proposes API-CD which shows strong performance by achieving 0% constraint violation rate.

2. In-context learning (ICL) is interesting approach to investigate for user utterances to API calls conversion.

3. Interesting constraints are specified which are related to the overall output of the task

4. The latest related work is referenced.

5. Error analysis of different constraint violations by different LLMs is also discussed.

6. Considering the in-context learning paradigm, this work is relevant.

**Reasons To Reject:**

1. Is the presented list of constraints exhaustive? This work is focused on hard constraints.

2. The proposed strategies are specifically designed for in-context learning and are not generalizable to other learning paradigms.

**Reproducibility:**

4: Could mostly reproduce the results, but there may be some variation because of sample variance or minor variations in their interpretation of the protocol or method.

**Reviewer Confidence:**

4: Quite sure. I tried to check the important points carefully. It's unlikely, though conceivable, that I missed something that should affect my ratings.

---

> ### Author Rebuttal · Authors · 2023-08-29
>
> We thank reviewer 13ew for their insightful reviews, and also for recognizing the value of our work on defining task-specific constraints violations, and our error analysis on existing LLMs.
>
> > Is the presented list of constraints exhaustive? This work is focused on hard constraints.
>
> In this work we focus on categorical constraints (or hard constraints). The presented list of constraints may not be exhaustive since we exclude the study of soft constraints (such as knowledge-driven constraints) from the scope of this work. However, we believe that the categorical constraints in this work are important in the context of executable semantic parsing, because these categorical constraints are crucial in ensuring that the generated outputs satisfy constraints from the pre-defined API and are therefore machine-interpretable (please also see the response to reviewer 56zM).
>
> > Weakness: The proposed strategies are specifically designed for in-context learning and are not generalizable to other learning paradigms.
>
> In this work we only focus on in-context learning, which is particularly appealing for the executable semantic parsing in practice. While we acknowledge that different learning paradigms may yield different outputs, we hope that future work can adopt our evaluation framework, which takes into consideration task-specific constraints when evaluating model outputs (please also see the response to reviewer 56zM).

---

### Official Review · Reviewer_56zM · 2023-08-10

**Soundness:** 3

**Excitement:**

3: Ambivalent: It has merits (e.g., it reports state-of-the-art results, the idea is nice), but there are key weaknesses (e.g., it describes incremental work), and it can significantly benefit from another round of revision. However, I won't object to accepting it if my co-reviewers champion it.

**Paper Topic And Main Contributions:**

This paper focuses on measuring and addressing constraint violations in in-context learning for generating task-oriented semantic parsing utterance-to-API calls.
1. Categorizing constraint types (structural, functional, argument, function-argument association) and creating metrics to gauge violations in generated API calls.
2. Studying constraint violations in leading language models, highlighting issues like inadequate relevant prompts and challenges in handling compositionality.
3. Introducing two solutions: semantic retrieval of demonstrations (SRD) for prompt formulation and API-aware constrained decoding (API-CD) during generation.
4. Demonstrating that SRD and API-CD effectively reduce constraint violations, with the best outcomes arising from their combined use. The study also examines tradeoffs between complexity, efficiency, and effectiveness of these solutions.

**Questions For The Authors:**





**Reasons To Accept:**

Addresses an important issue in applying in-context learning for executable semantic parsing - ensuring generated API calls respect task constraints and specifications. This is crucial for real-world deployment.
Provides useful analysis and insights into why constraint violations occur in state-of-the-art LLMs using in-context learning.
The proposed metrics, analysis, and mitigation strategies could aid progress in applying LLMs to executable semantic parsing and other code/program generation tasks.

**Reasons To Reject:**

The scope is quite narrow - focused solely on in-context learning for one dataset. Results may not generalize to other datasets or different training methods.

The study focuses on categorical constraints, specifically those that can be automatically detected by comparing generated API calls with predefined dictionaries, while excluding knowledge-driven constraints

**Reproducibility:**

4: Could mostly reproduce the results, but there may be some variation because of sample variance or minor variations in their interpretation of the protocol or method.

**Reviewer Confidence:**

3: Pretty sure, but there's a chance I missed something. Although I have a good feel for this area in general, I did not carefully check the paper's details, e.g., the math, experimental design, or novelty.

---

> ### Author Rebuttal · Authors · 2023-08-29
>
> We thank reviewer 56zM for their insightful review!
>
> We appreciate reviewer 56zM’s recognition of the value of our work in defining, analyzing and mitigating task-specific constraints violations in executable semantic parsing, which is not studied in previous works.
>
> > Weakness 1: This work only focused on in-context learning for one dataset. Results may not generalize to other datasets or different training methods.
>
> **In-context learning paradigm**:
> In the scope of this work, we only focus on in-context learning. We acknowledge that other alternative learning paradigms may yield different outcomes. However, the in-context learning paradigm holds substantial importance and practical appeals in the context of executable semantic parsing because it facilitates transfer learning in low-resource semantic parsing scenarios and enables rapid prototyping without the need of re-training. Even though our emphasis is on in-context learning in this work, we hope our fine-grained evaluation framework of constraints violation analysis can inspire future research on executable semantic parsing, to **study the models’ ability to meet pre-specified constraints**, going beyond only relying on conventional evaluation metrics such as exact match.
>
> **Dataset**:
> We use the TopV2 dataset since it is a popular task-oriented semantic parsing dataset, targeting the low-resource and few-shot learning setting. Also it can be readily converted to the utterance-to-API generation format. While an alternative viable dataset is presented by Shu et al 2022, we note that this dataset is small in size and not released yet.
>
> > Weakness 2: The study focuses on categorical constraints, specifically those that can be automatically detected by comparing generated API calls with predefined dictionaries, while excluding knowledge-driven constraints
>
> We limit the focus of this work to categorical constraints (or hard constraints, where for each constraint satisfaction condition C, C is 0 or 1). Categorical constraints are crucial in executable semantic parsing since the generated output often needs to be machine-interpretable or executable according to predefined APIs.
> We exclude the knowledge-driven constraints (soft constraints) from the scope of this study,  since studying such soft constraints may require more carefully curated datasets and benchmarks. We hope our work can inspire further investigations to follow our framework of formulating task-specific constraints-based metrics, rather than relying only on traditional string-match based metrics.
>
> ### Reference:
> Dialog2API: Task-Oriented Dialogue with API Description and Example Programs. Shu et al, 2022. arXiv:2212.09946

---

### Meta-Review · Area_Chair_2vZq · 2023-09-14

**Recommendation:** 3

**Metareview:**

The paper proposes metrics and subsequent mitigation strategies for constraint violations during utterance-to-api semantic parsing, focusing on LLMs and the in-context learning paradigm.

**Pros**: Reviewers agree on the relevance of the problem, and further, agree the paper provides valuable insights to the community, citing the proposed metrics, analysis, and mitigation strategies as having the potential to advance research.

**Cons**: Reviewers agree on a lack of generalizability and limited scope, citing limited datasets and application to only one learning paradigm. Author's rebuttal emphasizes a lack of available resources, and the importance of this particular paradigm, respectively. Reviewers acknowledge this argument, but the concerns remain. In the end, no reviewer finds the work particularly exciting - there is no champion.

---

### Decision · Program_Chairs · 2023-10-07

**Decision:**

Accept-Findings

**Comment:**

The paper proposes metrics and subsequent mitigation strategies for constraint violations during utterance-to-api semantic parsing, focusing on LLMs and the in-context learning paradigm.

**Pros**: Reviewers agree on the relevance of the problem, and further, agree the paper provides valuable insights to the community, citing the proposed metrics, analysis, and mitigation strategies as having the potential to advance research.

**Cons**: Reviewers agree on a lack of generalizability and limited scope, citing limited datasets and application to only one learning paradigm. Author's rebuttal emphasizes a lack of available resources, and the importance of this particular paradigm, respectively. Reviewers acknowledge this argument, but the concerns remain. In the end, no reviewer finds the work particularly exciting - there is no champion.